# Biomedical Advances in ABCA1 Transporter: From Bench to Bedside

**DOI:** 10.3390/biomedicines11020561

**Published:** 2023-02-15

**Authors:** Hong Y. Choi, Senna Choi, Iulia Iatan, Isabelle Ruel, Jacques Genest

**Affiliations:** 1Research Institute of the McGill University Health Centre, Montréal, QC H4A 3J1, Canada; 2Centre for Heart Lung Innovation, Department of Medicine, St. Paul’s Hospital, University of British Columbia, Vancouver, BC V6Z 1Y6, Canada

**Keywords:** ABCA1, atherosclerosis, cholesterol, high-density lipoprotein, desmocollin 1, docetaxel

## Abstract

ATP-binding cassette transporter A1 (ABCA1) has been identified as the molecular defect in Tangier disease. It is biochemically characterized by absence of high-density lipoprotein cholesterol (HDL-C) in the circulation, resulting in the accumulation of cholesterol in lymphoid tissues. Accumulation of cholesterol in arteries is an underlying cause of atherosclerosis, and HDL-C levels are inversely associated with the presence of atherosclerotic cardiovascular disease (ASCVD). ABCA1 increases HDL-C levels by driving the generation of new HDL particles in cells, and cellular cholesterol is removed in the process of HDL generation. Therefore, pharmacological strategies that promote the HDL biogenic process by increasing ABCA1 expression and activity have been intensively studied to reduce ASCVD. Many ABCA1-upregulating agents have been developed, and some have shown promising effects in pre-clinical studies, but no clinical trials have met success yet. ABCA1 has long been an attractive drug target, but the failed clinical trials have indicated the difficulty of therapeutic upregulation of ABCA1, as well as driving us to: improve our understanding of the ABCA1 regulatory system; to develop more specific and sophisticated strategies to upregulate ABCA1 expression; and to search for novel druggable targets in the ABCA1-dependent HDL biogenic process. In this review, we discuss the beginning, recent advances, challenges and future directions in ABCA1 research aimed at developing ABCA1-directed therapies for ASCVD.

## 1. Brief History of ABCA1, Tangier Disease

The cholesterol molecule evolved through the eons of evolution to maintain proper cell function [1]. Cellular cholesterol homeostasis is exquisitely regulated by multiple, redundant systems that ensure a narrow concentration range [2]. Excess cholesterol leads to cell autophagy and apoptosis through the unfolded protein response [3]. Some cells, especially M_2_ macrophages, have developed the ability to esterify cholesterol and store it in cholesteryl ester vesicles for transport to the sites of cholesterol catabolism, such as the spleen in mammals. However, when this system is overwhelmed by an abundant availability of oxidized low-density lipoprotein cholesterol (LDL-C), it can lead to the formation of foam cells, one of the earliest pathological findings in atherosclerosis. The intracellular regulatory pathway for cholesterol has been reviewed elsewhere [2] and will not be further dealt with here. The removal of cholesterol from cells by apolipoprotein A-I (apoA-I) is another mechanism by which accumulated cellular cholesterol can egress [4].

An interesting patient phenotype was identified in a young boy from the island of Tangier in the Chesapeake Bay, Maryland in the early 1960s. The patient had enlarged tonsils, hepato-splenomegaly, and was referred to the burgeoning National Institutes of Health (NIH). On extensive phenotyping, he was found to have an absence of high-density lipoprotein cholesterol (HDL-C). This new disease, named Tangier disease, became the topic of intense research since. The cellular defect was later identified to be an inability of apoA-I to promote cellular cholesterol efflux [5]. Family studies located the defect on chromosome 9q22-31. Multiple lines of evidence indicated that Tangier disease was caused by a cellular defect in removing cholesterol by apoA-I. The term used, cellular cholesterol efflux defect, became the main focus of research into its etiology.

The adenosine triphosphate (ATP)-binding cassette transporter A1 (ABCA1) was identified by Luciani et al. over 36 years ago by PCR cloning and found to be located on chromosome 9q22-31 [6]. ABCA1 belongs to what was then a growing family of transmembrane proteins sharing many structural and functional similarities. ABCA1 was initially thought to be involved in the phagocytosis of apoptotic cells and to play a role in the regulation of the inflammatory response [7,8]. Several groups simultaneously reported that pathogenic variants in the *ABCA1* gene were the molecular basis of Tangier disease, a recessive orphan disease characterized clinically by hepato-splenomegaly, enlarged tonsils, described pathologically as lipid-laden macrophages and lymphoid tissues, a progressive peripheral neuropathy, and represented biochemically by an absence of HDL-C in the blood [9,10,11,12,13].

An inverse relationship between HDL-C levels and the presence of atherosclerotic cardiovascular disease (ASCVD) was identified in multiple epidemiological studies. This association is strong, graded, and coherent across populations studied. HDL-C became a potential therapeutic target, and the search was on to identify compounds that would promote cellular cholesterol efflux via the ABCA1 pathway [14,15]. This relationship, between HDL-C, ABCA1 genetic variants, and ASCVD was challenged by Mendelian randomization studies showing that hypomorphic variants at the *ABCA1* gene that cause a low HDL-C did not increase the risk of ASCVD [16] and has been confirmed in large-scale studies. The failure of drugs that raise HDL-C, especially inhibitors of the cholesteryl ester transfer protein [17] and fibric acid derivatives, inhibitors of the peroxisome proliferator-activated receptor alpha (PPARα) has led to a decreased enthusiasm in seeking novel drugs that raise HDL-C [18].

The static measurement of the cholesterol content of HDL, however, does not necessarily reflect the dynamic lipid transfer between cells, especially macrophages and the liver, the “reverse cholesterol transport”. The rate-limiting step of cellular cholesterol efflux is considered to be the ABCA1-mediated transfer of cellular cholesterol to lipid-poor apoA-I. This can be measured in vitro as the cholesterol efflux capacity, and has shown promise as a predictive biomarker of cardiovascular disease [19,20,21].

This review will predominantly focus on the role of ABCA1 in cellular cholesterol efflux and describe the structure, mechanism of action, regulation of ABCA1, as well as new findings in the ABCA1 pathway. 

## 2. The Structure and Function of ABCA1

The ATP-binding cassette (ABC) transporters are a superfamily of transmembrane proteins that utilize ATP to transport a variety of substrates across cellular membranes. The ABC-A subfamily comprises 12 members, ABCA1 to ABCA13, with ABCA11 representing a transcribed pseudogene and is mostly related with lipid trafficking [22]. ABCA1 drives cholesterol efflux by exporting cholesterol and phospholipids (PLs) across the plasma membrane (PM) [23] and is a full-size ABC transporter containing two extracellular domains (ECD1 and ECD2), two transmembrane domains (TMD1 and TMD2), two nucleotide-binding domains (NBD1 and NBD2), and two regulatory domains (RD1 and RD2) (Figure 1).

The human ABCA1 protein is composed of 2261 amino acids and its structure was solved by single-particle cryo-electron microscopy (cryo-EM) in 2017 with resolutions of 3.9–4.1 Å (Figure 1A) [24]. The cryo-EM structure has revealed that ABCA1 is different from other ABC transporters in several features. First, ABC transporters typically form a cavity between the two TMDs to bind and translocate their substrates across membranes; however, the TMD1 and TMD2 in ABCA1 contact partially and do not form a sealed cavity, leading to exposure of the TMD side surfaces to the lipid bilayer. This suggests that ABCA1 may allow lateral entry of lipid substrates from the lipid bilayer. Second, it has been generally accepted that the alternation between inward- and outward-facing TMD conformations makes it possible for ABC exporters to translocate their substrates across membranes [23,25]. In the alternating access model, the TMD in ATP-free ABC exporters is opened toward the cytoplasm (inward-facing conformation), while the TMD in ATP-bound ABC exporters is opened toward the extracellular side (outward-facing conformation). As seen in Figure 1A,B, the TMD in ATP-free ABCA1 is in an outward-facing conformation, implying that the alternating access model may not be compatible with the ATP-dependent lipid flopping activity of ABCA1. Third, the ABC-A subfamily members, including ABCA1, are differentiated from most other ABC transporters due to the presence of large ECDs. The discovery of the ABCA1 structure has revealed that the distinguished ECDs in ABCA1 enclose a hydrophobic tunnel, ~60 Å in length. The PM is estimated to be 70–100 Å thick and composed primarily of a PL bilayer; therefore, the length of a single PL molecule may range from 35 to 50 Å, suggesting that the hydrophobic tunnel may serve as a passageway for the delivery of PLs. Based on the structural analysis, two ABCA1-mediated PL export mechanisms have been proposed (Figure 1A): (a) PLs in the inner leaflet of the PM enter into the interface between TMD1 and TMD2, subsequently they are flopped into the outward-facing transmembrane cavity, and then translocated into the hydrophobic tunnel [24]; (b) PLs in the outer leaflet of the PM diffuse into the outward-facing transmembrane cavity, and then they are translocated into the hydrophobic tunnel [26]. Both of the mechanisms are distinct from the mechanisms reported for the other ABC exporters.

The cryo-EM structure of human ABCA4 was determined in 2021 with resolutions of 3.3–3.4 Å [27,28]. ABCA4 is one of the closest relatives of ABCA1, and the comparison of the two cryo-EM structures has exhibited that the general architecture of ABCA1 and ABCA4 in the ATP-free state is very similar, but that the cytoplasmic RDs of the two are substantially different [28,29]. The RDs in ABCA4 cross over the axis of symmetry to form a domain-swapped pair of latches, while the ABCA1 RDs seen in Figure 1A have been modeled to make simple contacts without domain-swapping [28,29]. At the amino acid level, the RDs in human ABCA1 and ABCA4 show 59% identity and 76% similarity. This high degree of sequence homology and the domain-swapped latches unveiled in the higher-resolution structure of ABCA4 suggest that the ABCA1 RDs may also form domain-swapped latches. Aller and Segrest carefully inspected the original cryo-EM density map of ABCA1 and have proposed an alternative structure of human ABCA1 that contains domain-swapped RDs (Figure 1B) [29]. These RDs function like latches, so they constrain the NBDs to hold them in close proximity even in the absence of ATP binding. The authors have speculated that the swapping of RDs may be conserved in the entire ABC-A subfamily, based on their amino acid sequence analysis [29]. This swapped architecture may also be important for the unique action mechanisms of the ABC-A subfamily in comparison to the other subfamilies. 

Sun and Li solved the cryo-EM structures of human ABCA1 in both ATP-free and ATP-bound states in 2022 at 3.1 Å resolution [30]. These structures have exhibited that the ABCA1 RDs form domain-swapped latches. More interestingly, these structures have revealed the presence of a sterol-like molecule in the ECDs, and the authors have compared ATP-free with ATP-bound structures to propose an ABCA1-mediated cholesterol export model (Figure 1C). First, in the ATP-free state, cholesterol is loaded into the inner leaflet space between the TMDs, where a putative cholesterol-binding site is located. Next, ATP binding to the NBDs pulls the NBDs closer together, stretching the TMDs, inducing conformational changes in the TMD-ECD interface, leading to a counter-clockwise rotation of the ECDs by approximately 30 degrees. The authors have suggested that the closure of the outward-facing transmembrane cavity during the TMD stretching, plus the significantly changed TMD-ECD interface conformation, may drive the translocation of cholesterol from the inner leaflet space into the hydrophobic tunnel. They speculated that the translocation of cholesterol from the outer membrane leaflet into the hydrophobic tunnel might also be possible. Finally, upon release of ADP (the product of ATP hydrolysis), the NBD, TMD, and ECD conformations return to their ATP-free states.

**Figure 1 biomedicines-11-00561-f001:**
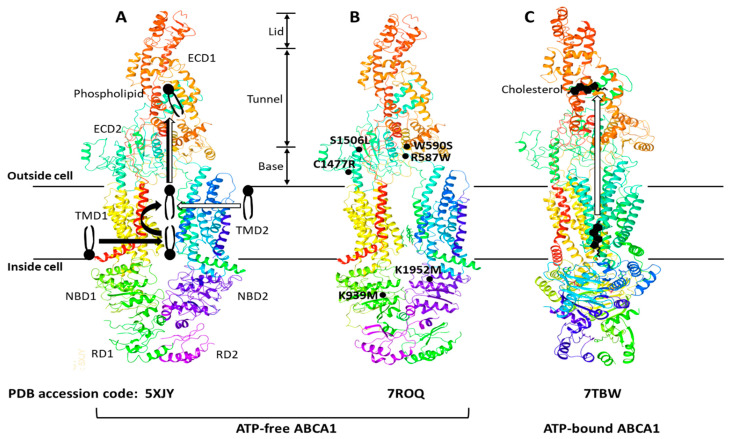
**ABCA1 structures, proposed lipid transport mechanisms, and hypomorphic missense mutations.** The ABCA1 comprises two extracellular domains (ECD1 and ECD2), two transmembrane domains (TMD1 and TMD2), two nucleotide-binding domains (NBD1 and NBD2), and two regulatory domains (RD1 and RD2). The ECD1 and ECD2 are intertwined, and the ECD can be subdivided into three parts—lid, hydrophobic tunnel, and base. The TMD1 and TMD2, each consisting of six transmembrane amphipathic α-helices, are separately folded. The RDs form a structural latch that may stabilize the NBDs and support effective cooperation between the two halves of ABCA1. Three ABCA1 structures—5XJY, 7ROQ, and 7TBW—were retrieved from the protein data bank (PDB) (https://www.rcsb.org) (accessed on 13 December 2022). Two different mechanisms of phospholipid export have been proposed with the initial structure 5XJY, and the mechanisms are depicted with thick black and white arrows in (**A**) [24,26]. The initial structure was revised to propose an alternative structure 7ROQ seen in (**B**) [29]. The RD1 and RD2 in (**B**) are swapped, and each RD forms a domain-swapped latch. The locations of representative hypomorphic missense mutations are shown in (**B**): R587W and S1506L variants are impaired in localization at the plasma membrane; W590S is defective in phospholipid translocation activity; C1477R reduces cellular apoA-I binding; K939M and K1952M disrupt ATPase activity [31]. A cholesterol export mechanism has been proposed with the 7TBW seen in (**C**) [30].

These studies investigating structure-function relationships have proposed plausible mechanisms for ABCA1-mediated transport of PLs and cholesterol across the PM. The structural-based functional analysis has provided new insights into the ABCA1 action mechanisms and has also raised new questions: How are PLs and cholesterol loaded into the space between the TMDs? Are both PLs and cholesterol direct substrates of ABCA1? If they are, how does ABCA1 achieve the dual-substrate specificity? How are PLs and cholesterol in the hydrophobic tunnel transferred to apoA-I? Further studies are warranted for us to fully understand the ABCA1-mediated lipid export. 

## 3. ABCA1 and Biogenesis of HDL

The ABCA1-mediated transfer of cellular lipids to extracellular lipid-poor apoA-I generates lipid-apoA-I complexes called nascent HDL particles. This process is termed HDL biogenesis and occurs through complex interactions between ABCA1, lipids, and apoA-I in the PM. HDL biogenesis has still been incompletely understood, and nine different models have been put forth for the HDL biogenic process [32].

There are technical limitations in investigating ABCA1-lipids-apoA-I interactions. Most of all, it is difficult to directly measure lipid-protein interactions at the molecular level under physiologically relevant conditions. High-resolution structural methods, such as X-ray crystallography and cryo-EM, have demonstrated their capability in visualizing lipid-protein interactions directly. It is however rare for the structural methods to visualize endogenous lipid molecules bound to proteins. The conformational/compositional heterogeneities and molecular dynamics of lipid-protein complexes make it challenging to obtain their ordered crystallization or build computational models into cryo-EM density maps. In addition to the interactions between ABCA1, lipids, and apoA-I, the composition and biophysical properties of the PM that facilitate HDL biogenesis should be investigated. The PM consists of a mosaic of functional microdomains, and the identification of PM microdomains associated with HDL biogenesis may shed light on elucidating the HDL biogenic process.

There is no consensus model that can reconcile the differences between the nine HDL biogenesis models, so we need to build a plausible model that is supported by major findings. The recent structure-based functional analysis has suggested that ABCA1 translocates both PLs and cholesterol from the PM to the ECD of ABCA1 (Figure 1). This ABCA1 function was proposed by one of the nine models [32]. In the model, ABCA1 monomers sequester cholesterol and PLs within their ECDs and undergo conformational changes to dimerize once a sufficient amount of the lipids is sequestered [33,34]. This model has suggested that the ECD may function as a temporary lipid reservoir. However, ABCA1 structures show that the ECD forms a narrow elongated ~60 Å tunnel [24], suggesting that the ECD may function as a passageway for lipid transport rather than a wide lipid reservoir. There are two main methods that explain the transfer of the lipids in the ECD to apoA-I [32]: (a) a direct transfer via ABCA1-apoA-I interactions; (b) an indirect transfer via the formation of a PM microdomain on the outer leaflet. In the second method, it is thought that the lipids in the ECD are loaded into the microdomain to facilitate lipid-apoA-I interactions. Cross-linking and binding studies have shown that ABCA1-apoA-I interactions account for only approximately 10% of apoA-I binding to cells, whereas lipid-apoA-I interactions in a PM microdomain created by ABCA1 account for most of apoA-I binding [35]. Computational simulations have shown that the PL translocase activity of ABCA1 increases PL densities in the outer leaflet, therefore pressurizing the leaflet to create an exovesiculated PM microdomain [36]. Amphipathic α-helices of apoA-I have intrinsic lipid binding properties and bind to the PM microdomain [23]. Based on these observations, we propose a plausible model: ABCA1 translocates PLs and cholesterol through the hydrophobic tunnel to the cell surface; the lipids then create a microdomain on the outer PM leaflet by unknown mechanisms; the microdomain is next wrapped by apoA-I to produce nascent HDL particles. Our simplified HDL biogenesis model is shown in Figure 2.

## 4. Regulation of ABCA1

The essential role of ABCA1 in HDL biogenesis has indisputably demonstrated that upregulation of ABCA1 expression and activity will promote HDL biogenesis. All ABCA1-expressing cells can contribute to HDL biogenesis, and hepatic and intestinal HDL biogenesis are the major sources of plasma HDL [37,38]. To develop therapeutic strategies for raising HDL biogenesis, the molecular mechanisms of ABCA1 expression/activity regulation have been extensively studied for more than two decades. 

ABCA1 was initially identified as a sterol-sensitive gene with binding sites for the nuclear receptors liver-X-receptor (LXR) and retinoid-X-receptor (RXR) in the ABCA1 promoter. Nuclear receptors are ligand-activated transcription factors and the LXR/RXR heterodimer binding site has been mapped to the direct repeat 4 (DR4) element in the ABCA1 promoter. Follow-up studies have shown that other nuclear receptors such as PPARs also regulate the transcription of ABCA1 via the DR4 element. These nuclear receptors are major positive regulators of ABCA1 gene transcription. Therefore, activation of these receptors has been a main strategy for upregulation of ABCA1 expression [39,40]. In addition to conventional transcription factors, epigenetic modifications including histone acetylation and DNA methylation emerge as new regulators of ABCA1 gene transcription [41,42].

Micro RNAs (miRNA) are single-stranded, short (~22 nucleotides) non-coding RNA molecules and have been identified as key regulators of ABCA1 expression at the post-transcriptional level, by pairing with the ABCA1 (or other targets) mRNA and causing its degradation. In most cases, miRNA suppress gene expression. These include miRNA-9-5p, miRNA-10b, miRNA-19b, miRNA-20a/b, miRNA-23a-5p, miRNA-26a/b, miRNA27a/b, miRNA-33a/b, miRNA-106b, miRNA-128-2, miRNA-144, miRNA-145, miRNA-148a, miRNA-183, miRNA-613, and miRNA-758 [39,43]. Many long non-coding RNAs (lncRNAs, RNA transcripts >200 nucleotides), including MeXis, GAS5, TUG1, MEG3, MALAT1, Lnc-HC, RP5-833A20.1, LOXL1-AS1, CHROME, DAPK1-IT1, SIRT1 AS lncRNA, DYNLRB2-2, DANCR, LeXis, LOC286367, and LncOR13C9, are also involved in the regulation of ABCA1 expression [44]. This long list of non-coding RNAs brings us to a set of interesting but challenging questions: Why are so many involved in regulating ABCA1? Do some of them function synergistically or antagonistically? Do they make an ABCA1 regulatory network? Can we break the network by targeting a few of them? Once these questions beget answers, druggable non-coding RNA targets might be represented.

The cellular levels of ABCA1 protein are largely regulated by calpain-, lysosome- and proteasome-mediated degradation pathways [45,46,47]. It has been reported that pharmacological inhibition of ABCA1 degradation increases HDL biogenesis [48], but we do not fully understand how the protein degradation pathways select their substrates, which makes it difficult to specifically inhibit ABCA1 degradation out of the numerous proteins targeted. 

These complicated regulation systems suggest that tight regulation of ABCA1 expression may be required to maintain cellular lipid homeostasis. 

## 5. Translational Biology and Drug Trials

Small molecules that modulate ABCA1 expression have been tested for their ability to increase HDL biogenesis and promote reverse cholesterol transport. This became a test of the “HDL hypothesis” that stipulates that raising HDL will be beneficial to prevent and treat ASCVD.

The identification of certain oxysterols and 9-cis-retinoic acid as ligands for LXR and RXR, respectively, triggered a grand race for developing LXR and RXR agonists, and many potent agonists have been developed. The effects of LXR and RXR agonists on the upregulation of ABCA1 expression in vitro studies raised hope that the agonists could be used for therapeutic purposes. However, none of them has reached the clinic, due to adverse side effects such as hypertriglyceridemia and hepatic steatosis [43].

Niacin, also known as nicotinic acid, induces ABCA1 expression via a DR4 element-dependent mechanism, as well as increasing LXR expression and apoA-I levels. Niacin was used for the treatment of dyslipidemias in the pre-statin era, and two large clinical trials examined the effects of niacin on cardiovascular risk. The Atherothrombosis Intervention in Metabolic Syndrome with Low HDL/High Triglycerides: Impact on Global Health Outcomes (AIM-HIGH) trial randomized 3414 patients to receive niacin or placebo. Despite a significant increase in HDL-C levels, the trial was stopped after a mean follow-up period of 3 years owing to a lack of efficacy [49]. In the Heart Protection study-2 (HPS2-THRIVE), 42,424 patients with occlusive ASCVD were given simvastatin 40 mg (with ezetimibe, if required) and randomized to receive extended-release niacin with laropiprant to prevent the cutaneous flushing associated with niacin. Adding niacin to statin-treated patients did not reduce the risk of major adverse cardiovascular events but increased the risk of serious adverse events [50].

The PPARα agonists comprising the fibric acid derivatives (fenofibrate, bezafibrate, gemfibrozil, clofibrate, and pemafibrate) are predominantly used for the treatment of elevated plasma triglyceride levels, and the PPARγ agonists (glitazones) are used for the treatment of diabetes. These drugs have been extensively tested clinically for their ability to increase ABCA1 and HDL-C levels [43]. These fibric acid derivatives are potent and selective synthetic agonists of PPARα. Their main effect is to lower plasma triglyceride levels by increasing the activity of lipoprotein lipase, predominantly through inhibition of apo CIII secretion and increased secretion of apo A-V, an activator of lipoprotein lipase. Selective PPARα agonists and pan PPAR agonists such as bezafibrate also upregulate the transcription of ABCA1 and have been shown to increase cellular cholesterol efflux in a variety of human cell lines. Because of the multiple effects of PPARα agonists on human lipoprotein metabolism, the absolute contribution of PPARα agonists to ABCA1-mediated cholesterol efflux and plasma HDL-C levels remains uncertain. Two trials of fenofibrate have tested the hypothesis that lowering triglyceride levels in patients at high cardiovascular risk would have a beneficial effect on cardiovascular outcomes. The Fenofibrate Intervention and Event Lowering in Diabetes (FIELD) trial tested fenofibrate in 9795 patients with type 2 diabetes. Plasma levels of HDL-C did not increase significantly at the end of the study. Fenofibrate did not significantly reduce the risk of the primary outcome of coronary events [51]. In this study, the extensive use of statins likely masked a potential benefit of fenofibrate. In the Action to Control Cardiovascular Risk in Diabetes (ACCORD) trial, 5518 patients with type 2 diabetes, treated with open-label simvastatin, were randomized to fenofibrate or placebo. Mean HDL cholesterol levels increased from 0.98 to 1.07 mmol/L (38.0 to 41.2 mg/dL) in the fenofibrate group and from 0.99 to 1.05 mmol/L (38.2 to 40.5 mg/dL) in the placebo group. The addition of fenofibrate to simvastatin-treated patients did not reduce the rate of fatal cardiovascular events, nonfatal myocardial infarction, or nonfatal stroke [52]. 

Pooled sub-group analysis of trials with fibric acid derivatives suggested a potential benefit in patients with residual hypertriglyceridemia with low HDL-C [53]. In order to test this hypothesis, the Pemafibrate to Reduce Cardiovascular Outcomes by Reducing Triglycerides in Patients with Diabetes (PROMINENT) trial using pemafibrate examined 10,497 patients with type 2 diabetes, triglyceride levels between 2.3 and 5.6 mmol/dL (200 and 499 mg/dL), and HDL-C levels < 1.0 mmol/L (40 mg/dL). Despite a favorable change in atherogenic lipoprotein lipid profile, there was no change in HDL-C levels. The pre-specified primary end-point event occurred in 572 patients in the pemafibrate group and in 560 of those in the placebo group (HR 1.03; 95% CI, 0.91 to 1.15) [54].

The PPARγ agonists pioglitazone, rosiglitazone, and troglitazone have been used for the treatment of type 2 diabetes. These agonists have also been shown to increase ABCA1 expression. Pioglitazone was the most intensively studied for the prevention of major adverse cardiovascular events (MACE) and the development of heart failure. A meta-analysis of eight randomized clinical trials examined the effects of pioglitazone on MACE, heart failure, and all-cause mortality with 10,682 patients and 9674 controls. Pioglitazone did not reduce the risk of MACE, heart failure, or all-cause mortality [55].

Overall, these ABCA1 upregulating agents, especially the fibric acid derivatives frequently used in high-risk patients with combined lipoprotein disorders and diabetes, have failed to change clinical outcomes of cardiovascular death, non-fatal myocardial infarctions, or strokes. The large clinical trials are summarized in Table 1. Numerous naturally occurring compounds have also been examined for their ability to modulate ABCA1 expression. A number of natural compounds, such as retinoic acid, beta-carotene, and various herbal medicines, have been reported to increase ABCA1 expression in humans. However, these “nutraceuticals” have not been recommended as a means to increase HDL-C.

The cumulative data from clinical trials aiming to increase HDL-C through the manipulation of ABCA1 expression for the prevention of ASCVD have not met success, indicating that ABCA1 is an attractive yet elusive therapeutic target in ASCVD. 

## 6. A Novel Druggable Target in the ABCA1 Pathway

Although ABCA1 has been identified as a major target for promoting HDL functions and reducing the risk of ASCVD, extensive attempts to develop ABCA1-directed therapies for the past 24 years have proven the difficulty of modulating ABCA1 for therapeutic purposes. A multi-layered network consisting of transcriptional, post-transcriptional, translational, and post-translational regulations determines the expression level of ABCA1 [39,40,43]. The network is controlled by various signaling molecules such as hormones, lipid metabolites, and inflammatory mediators. These many different signals and regulatory steps are integrated for the production of proper amounts of ABCA1 protein. Besides, the ABCA1 protein turns over rapidly with a half-life of 1–2 h [56,57]. The multi-layered coordination and the short half-life have made it difficult to pharmacologically upregulate or activate ABCA1 without eliciting adverse side effects.

Moreover, there are several studies suggesting that upregulated ABCA1 protein levels are not necessarily associated with HDL biogenesis. Le Lay et al. [58] showed that ABCA1 expression was upregulated during the differentiation of 3T3-L1 preadipocytes to adipocytes, but ABCA1-dependent HDL biogenesis was not increased in differentiated adipocytes versus 3T3-L1 preadipocytes. The upregulation of ABCA1 expression was also observed during the differentiation of THP-1 monocytes to macrophages. The degree of THP-1 differentiation was positively correlated with ABCA1 expression levels, but negatively correlated with ABCA1-dependent HDL biogenesis [59]. These studies suggest that upregulation of ABCA1 may not be effective in promoting HDL biogenesis in differentiated adipocytes and macrophages. These cells possess a high capacity of storing cholesterol, and the storage capacity may limit the amount of cholesterol available for efflux by the ABCA1 pathway. It may therefore be necessary to redistribute cholesterol in storage pools to efflux-available pools for the stimulation of ABCA1 activity in the cells. Another cell type that is impaired in ABCA1-dependent HDL biogenesis is smooth muscle cells (SMCs) in the intimal layer of arteries: ABCA1 expression is reduced in intimal SMCs compared to medial arterial SMCs and intimal macrophages, and overexpression of ABCA1 in intimal-type SMCs fails to promote HDL biogenesis [60,61]. The results suggest that upregulation of ABCA1 expression may not be sufficient to correct the impaired HDL biogenesis, and that the impairment may contribute to the formation of cholesterol-laden SMC foam cells in the intima. Accumulation of cholesterol in SMC- and macrophage-derived foam cells in the arterial intima is the hallmark of atherogenesis, and HDL biogenesis is the primary mechanism to remove excess cholesterol from the foam cells. The lack of a positive correlation between ABCA1 expression levels and HDL biogenesis in differentiated macrophages and intimal-type SMCs has therefore suggested that ABCA1 might not be an ideal target for the promotion of atheroprotective HDL biogenesis. 

In addition to the difficulty of pharmacological modulation of ABCA1, the doubt on the validity of ABCA1 as a therapeutic target has urged searches for new targets that are druggable for the promotion of HDL biogenesis. To find such targets, we investigated PM microdomains involved in HDL biogenesis: it is believed that nascent HDL particles are formed by apoA-I-mediated solubilization of a PM microdomain created by ABCA1, thus we isolated PM microdomains and determined their molecular composition. In the course of the work, we purified a novel PM microdomain interacting with apoA-I [62]. Lipidomic and proteomic analysis revealed that the novel microdomain was rich in cholesterol and contained desmosomal proteins, followed by the identification of desmocollin 1 (DSC1) among these proteins as a new apoA-I binding protein. Functional studies have shown that DSC1 binds and sequesters apoA-I to prevent HDL biogenesis, which has led to the update of our HDL biogenesis model: ABCA1 upregulated in cholesterol-laden cells creates a PM microdomain for apoA-I to bind and solubilize the microdomain (HDL biogenesis), whereas DSC1 binds and sequesters apoA-I to protect cholesterol from being removed by the HDL biogenesis [62,63]. The identification of DSC1 as a negative regulator of HDL biogenesis has presented the first cell-surface protein target that can be inhibited for the promotion of HDL biogenesis. In drug discovery, targeting cell-surface proteins with specific inhibitors has been the most successful approach, which suggests that DSC1 may be a druggable target. 

## 7. Targeting of DSC1 with Docetaxel

The druggability of a protein is defined as the protein’s ability to bind drug-like molecules with high affinity. Computational assessments have predicted that the apoA-I binding site in DSC1 is druggable, and a structure-based virtual screening of 10 million small molecules has discovered several small molecules that have high affinity for the apoA-I binding site and promote HDL biogenesis [64]. The results suggest that inhibition of DSC1-apoA-I interactions may be a valid strategy for promoting HDL biogenesis. Among the high-affinity molecules, docetaxel (DTX) showed the highest potency in promoting HDL biogenesis; the half-maximal effective concentration of DTX in fibroblasts and SMCs was 0.72 nM and 0.35 nM, respectively [64]. The potency of DTX in macrophages was dependent on the degree of macrophage differentiation: the differentiation degree was inversely related with the expression level of DSC1; consequently, the amount of DTX required for inhibiting DSC1-apoA-I interactions was lower in highly differentiated macrophages than in poorly differentiated ones [59]. In highly differentiated ones, 1 nM of DTX was sufficient to significantly promote HDL biogenesis [59]. Our view of DTX-mediated promotion of HDL biogenesis is seen in Figure 2.

Differentiated macrophages and SMCs are the two most relevant cell types in the initiation and progression of atherosclerosis [61,65]. Cholesterol accumulation in these cells and their transformation into foam cells represent major atherogenic events; therefore, the HDL biogenic effect of DTX at low nanomolar concentrations in these cell types suggests that DTX may be an anti-atherogenic drug candidate. The low-nanomolar potency is particularly important as DTX has been approved by the FDA as a cancer chemotherapy drug. In chemotherapy, DTX has been used at the concentrations 1.9–5.1 μM in plasma [66,67], which is higher than HDL biogenic doses by at least three orders of magnitude. In an animal study using apoE null mice, we observed that treatment of the mice with DTX for 6 weeks significantly reduced atherosclerosis without causing hematologic and hepatic toxicity. During the treatment, the plasma concentrations of DTX were maintained in the range of 2.7–4.3 nM by using an osmotic drug delivery system (Choi HY et al., manuscript in preparation). DTX exerts its cytotoxic effects by binding to β-tubulin, a primary component of cytoplasmic microtubules [68,69]. To target the intracellular protein β-tubulin, DTX must first traverse the PM to reach the cytosol. The PM is selectively permeable; therefore, passive diffusion across the PM is usually limited to small molecule drugs. The PM barrier may limit DTX accessibility towards the intracellular protein β-tubulin and consequently increase the chance of binding to the extracellular DSC1, which may contribute to the markedly different potency of DTX: micromolar potency in targeting β-tubulin (chemotherapy) vs. nanomolar potency in targeting DSC1 (atheroprotection). For use of DTX as an atheroprotective drug, it will be important to optimize DTX formulation and dosage regimen not to build up cytotoxic concentrations during the DTX treatment. As DTX induces cell death by causing a cell-cycle arrest, highly proliferative cancer cells are more susceptible to DTX toxicity compared to normal, non-cancerous cells. For example, a chemotherapy dose of 5 μM induced cell death in cancer cells [70], whereas the viability of SMCs and macrophages was not affected until 25 μM DTX treatment [59,64]. These data suggest that nanomolar, non-cytotoxic concentrations of DTX may be used to reduce atherosclerosis.

The successful targeting of DSC1 with DTX has not only provided the opportunity to repurpose the chemotherapy drug DTX for use in reducing atherosclerosis, but has also opened the door to targeting DSC1-apoA-I interactions with other therapeutic agents, such as peptides and monoclonal antibodies. 

## 8. Conclusions

Since the discovery in 1999 that loss-of-function mutations in ABCA1 were associated with low HDL levels, ABCA1 has been studied as a promising therapeutic target for preventing and treating ASCVD. Structural, biochemical, and functional studies have shown that ABCA1 exports cellular PLs and cholesterol to apoA-I to generate HDL particles, and that upregulation of ABCA1 expression promotes the HDL biogenic process. As removal of excess cellular cholesterol via HDL biogenesis is believed to be the main atheroprotective function of ABCA1, therapeutic effects of ABCA1-upregulating agents in ASCVD have been highly sought after. However, large clinical trials with such agents have failed to improve clinical outcomes. The results have proven the difficulty of therapeutic upregulation of ABCA1, probably due to the complexity of the ABCA1 regulatory system. These trials have also been another example of demonstrating the difficulty of translating laboratory discoveries into patient care. To overcome the challenges in the development of ABCA1-directed therapies, extensive efforts to identify new druggable targets in the ABCA1 pathway have been made. The finding of DSC1 as a novel negative regulator of ABCA1-dependent HDL biogenesis and the identification of DTX as a potent inhibitor of the DSC1 action have opened a new horizon for developing ABCA1-directed therapies for ASCVD.

## Figures and Tables

**Figure 2 biomedicines-11-00561-f002:**
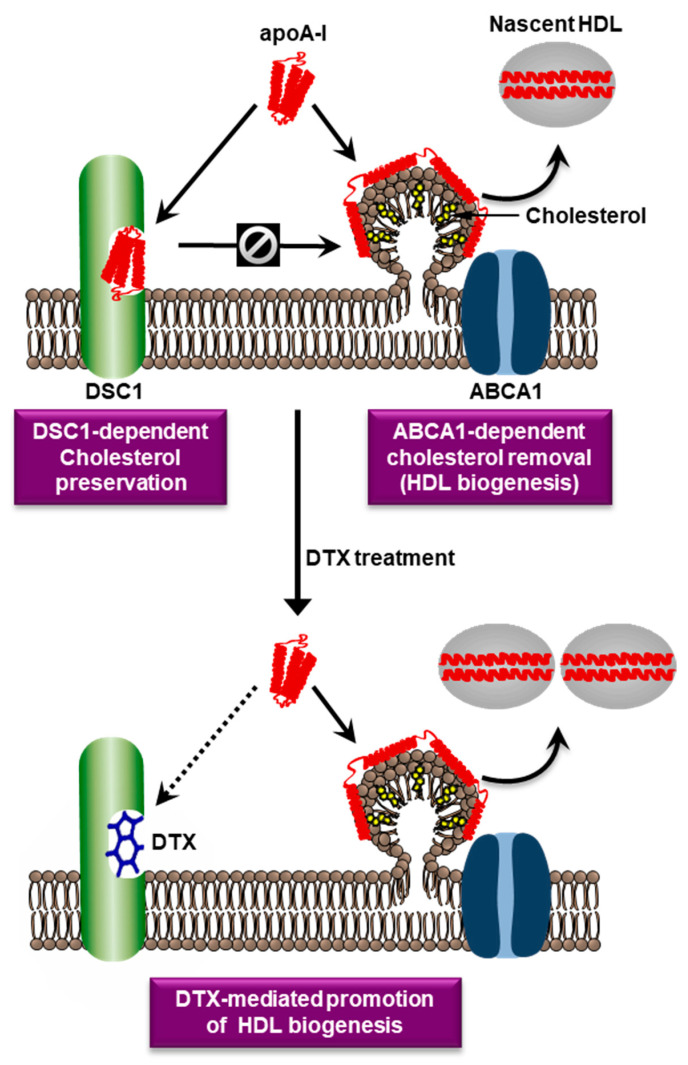
**Hypothetical roles of ABCA1, DSC1, and DTX in HDL biogenesis.** ABCA1 is upregulated in cholesterol-laden cells and creates a PM microdomain to remove excess cellular cholesterol. The microdomain surface may facilitate the binidng of an amphipathic protein apoA-I. Amphipathic α-helices of apoA-I possess detergent-like properties that can solubilize lipids in the microdomain. The apoA-I-lipid complexes dissociated from the microdomain are called nascent HDL particles. An apoA-I binding protein DSC1 sequesters apoA-I to prevent the ABCA1-dependent HDL biogenic pathway, which may be a cellular mechanim to preserve PM cholesterol levels. The two counteracting proteins ABCA1 and DSC1 may therefore be key determinants of HDL biogenesis. The blocking of the apoA-I binding site in DSC1 with DTX may increase apoA-I amounts that are available for the ABCA1-dependent HDL biogenic pathway.

**Table 1 biomedicines-11-00561-t001:** The results of large clinical trials investigating the effects of ABCA1 upregulating agents on ASCVD outcomes.

Class	Drug	Mechanism of Action	Clinical Trial	ASCVD Outcomes
Niacin	Niacin	LXR	AIM-HIGH [49], HPS2-THRIVE [50]	Neutral/Harm
Glitazones	Pioglitazone	PPARγ	Meta-analysis [55]	Neutral/Harm
Fibric acid derivatives	Fenofibrate, Pemafibrate	PPARα	FIELD [51], ACCORD [52], PROMINENT [54]	Neutral

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
