# Peer review of "Biomedical Advances in ABCA1 Transporter: From Bench to Bedside"

_biomedicines, 2023, doi:10.3390/biomedicines11020561_

Round 1

Reviewer 1 Report

H. Y. Choi et al presented a review in which they highlighted advances in preventing ASCVD by increasing the activity of the ABCA1 transporter. Based, among other things, on their own data, the authors presented the structure of the ABCA1 transporter, described its role in the synthesis of high-density lipoproteins, presented its mechanism of action, and gave an analysis of drug trials to reduce the risk of atherosclerotic events. In my opinion, the data presented in this review correspond to the profile of the journal Biomedicine. The review is a qualitative description of the data obtained so far. And it shows one of the promising directions in the search for means to prevent atherosclerotic phenomena, although only one of the existing possibilities.

As for the manuscript itself, I have only two remarks.

1. In the caption to Figure 1, the authors gave a transcript of all the abbreviations they used, except for RD. It would be logical in this case to use the full name, and then the abbreviation

2. An analysis of the literature shows that its bulk falls on the end of the last and the twenties of the current century. However, the first decade also saw a number of studies shedding light on both the causes of atherosclerosis and the role of the ABCA1 transporter and its encoding genes. In this review, the authors cite relatively few articles relating to 2004-2010, although they are of great importance for understanding the development of the disease and methods of its treatment. For example, the articles below. The authors are requested to consider the possibility of assessing the significance of these articles in understanding the situation as a whole.

Albrecht C, Viturro E. The ABCA subfamily--gene and protein structures, functions and associated hereditary diseases. Pflugers Arch. 2007 Feb;453(5):581-9. doi: 10.1007/s00424-006-0047-8. Epub 2006 Apr 4.

Brunham LR, Kruit JK, Iqbal J, Fievet C, Timmins JM, Pape TD, Coburn BA, Bissada N, Staels B, Groen AK, Hussain MM, Parks JS, Kuipers F, Hayden MR. Intestinal ABCA1 directly contributes to HDL biogenesis in vivo. J Clin Invest. 2006 Apr;116(4):1052-62. doi: 10.1172/JCI27352. Epub 2006 Mar 16.

Mott S, Yu L, Marcil M, Boucher B, Rondeau C, Genest J Jr. Decreased cellular cholesterol efflux is a common cause of familial hypoalphalipoproteinemia: role of the ABCA1 gene mutations. Atherosclerosis. 2000 Oct;152(2):457-68. doi: 10.1016/s0021-9150(99)00498-0.

Oram JF. HDL apolipoproteins and ABCA1: partners in the removal of excess cellular cholesterol. Arterioscler Thromb Vasc Biol. 2003 May 1;23(5):720-7. doi: 10.1161/01.ATV.0000054662.44688.9A. Epub 2003 Jan 9.

Tall AR, Yvan-Charvet L, Terasaka N, Pagler T, Wang N. HDL, ABC transporters, and cholesterol efflux: implications for the treatment of atherosclerosis. Cell Metab. 2008 May;7(5):365-75. doi: 10.1016/j.cmet.2008.03.001.

Tang C, Oram JF. The cell cholesterol exporter ABCA1 as a protector from cardiovascular disease and diabetes. Biochim Biophys Acta. 2009 Jul;1791(7):563-72. doi: 10.1016/j.bbalip.2009.03.011. Epub 2009 Apr 1

Author Response

We thank the Reviewer for the thoughtful comments, and here we enclose our response.

Reviewer 2 Report

This review article has attempted to collect current knowledge and advances, including authors' own research, regarding the efficacy of ABCA1-targeted drugs in the efflux and removal of intracellular cholesterol and the regulation of HDL-C biogenesis in the bloodstream to discuss the past, advances, and challenges of ABCA1 and ASCVD researches, hoping to provide future directions of developing ABCA1-engaged therapeutic strategies against ASCVD. Overall, this is a well-written review article in which the authors provide a concise but easily understandable background introduction about the initial studies of the rare genetically recessive Tangier disease with cellular cholesterol efflux defect and the therapeutic relationships/controversies among ABCA1, HDL-C, and ASCVD in basic and clinical studies. They exquisitely brief how cholesterol can structurally be transported from cytosol to the extracellular compartment where it can then be removed by binding to ApoA-I to increase the level of plasma HDL-C. They also detail the molecular regulation of the ABCA1 functionality and inhibitors used to target ABCA1-directed pathways in reducing ASCVD risks. Upon providing the clinical failures of these inhibitors in improving ASCVD-related problems, they point out possible future directions and design to identify potential and better drugs. Finally, they, based on their previous findings, also suggest a newly hypothesized strategy targeting DSC1 with Docetaxel for increasing HDL biogenesis, particularly in macrophages and SMCs most relevant to the initiation and progression of atherosclerosis, to reduce ASCVD risks. However, I do have two suggestions that may make the review more comprehensive.

1. Since ABCA-1 is such an essential molecule in understanding the molecular mechanism underlying HDL biogenesis defect and ASCVD etiology, the authors should briefly illustrate what are hypomorphic variants (what kinds of mutations) at the ABCA1 gene and how they structurally result in functional defects as to the cholesterol efflux.

2. Since Docetaxel (DTX) is a cancer chemo agent most well-known for its effects on microtubule polymerization within the cytosol compartment and drug-resistance to taxane treatment often due to enhanced activities of ABC family members that pump out the drugs, the authors should discuss how could this drug, instead of penetrating plasma membrane and entering cytosol, stay on the cell surfaces to bind to the extracellular domain of DSC1. Moreover, could it be possible that the effects of DTX on HDL biogenesis in fibroblasts and SMCs are due to their effects on microtubules instead of those directly on DSC1 functionalities?

Author Response

We thank the Reviewer for the insightful comments, and here we enclose our response. 

Reviewer 3 Report

In this review, Choi and colleagues try to describe the current knowledge of ABCA1 and its role in HDL formation. Furthermore, the authors discuss the failed attempts to upregulate the expression of ABCA1 pharmacologically. 

While I think this is an important topic that could trigger new research in the field, the review falls short in many parts by being to superficial and lacking original references. E.g. the introduction talks about the function of ABCA1 in cholesterol efflux but nothing is said about other lipids, and thus the statement in line 90 that ABCA1 translocates lipids across the plasma membrane without saying how is kind of irritating.

The structures shown in Figure 1 need a better description, especially concerning the translocation of phospholipids and cholesterol. What I read in the text I do not see in the Figures. Especially, lines 147 – 158 are very difficult to follow with the given Figure. 

Lines 186/187: What type or kind of 9 models? Does one of them make more sense with the given structures?

In Figure 2 the DSC1 and DTX should be explained because the figure is mentioned way before DSC1 and DTX are explained in the text. Also where in the figure is “nascent HDL” that is mentioned in the legend?

The regulation paragraph is not very clear and perhaps summarizing the different miRNAs and lncRNAs in a Table might help. Also, most of the references cited are other reviews which is not very helpful if I want to see the original research that is described.

In line 295 the pharmacological inhibition requires a reference.

The section on translational biology and drug trials again should have more original references rather than reviews.

Table 1 does not really help. There were only a few trials and they were easy to follow in the text.

The first part of the section on novel findings belongs to the section on regulation. It looks like different authors contributed to the different sections without reading other parts of the review. Also, I do not really see novel findings in the novel findings section; with the exception of the 2022 paper by the authors, the other papers are at least 6 years old and many even older (2002 – 2004, 2009). Please better emphasize what the novel findings are.

Author Response

We thank the Reviewer for the constructive comments, and here we enclose our response.

Round 2

Reviewer 2 Report

The authors have satisfactorily addressed my concerns and revised the manuscript accordingly. I have no further questions.

Reviewer 3 Report

The authors have addressed my comments and suggestions, and I am fine with keeping the tables and references based on their explanations.